# Strength Estimation and Fundamental Characteristics of the New Rotational Friction Damper with Translational Movement

Shintaro Tahara [1], Koshiro Iwaya [1], Tsutomu Iwashita [2,*], Katsuhiko Goto [3] and Minoru Yamanari [4]

[1] Advanced Engineering Course, National Institute of Technology, Ariake College, Omuta 836-8585, Fukuoka, Japan; s55109@ga.ariake-nct.ac.jp (S.T.); s56018@ga.ariake-nct.ac.jp (K.I.)
[2] Architecture Course, National Institute of Technology, Ariake College, Omuta 836-8585, Fukuoka, Japan
[3] Department of Architecture and Civil Engineering, National Institute of Technology, Kumamoto College, Koshi 861-1102, Kumamoto, Japan; goto@kumamoto-nct.ac.jp
[4] Former Professor, Kumamoto University, Kumamoto 860-8555, Kumamoto, Japan; yamanari@arch.kumamoto-u.ac.jp
[*] Correspondence: iwashita@ariake-nct.ac.jp

**Abstract:** This paper explores the strength estimation and hysteretic characteristics of a rotational friction damper with translational movement. The experiment comprised two distinct cases: one involving only rotation and the other allowing rotation combined with translational movement. Notably, in the case integrating translational movement, the load demonstrated an incremental trend aligned with displacement, attributing this behavior to the effect of translational movement. Two strength estimation models were introduced and subsequently benchmarked against both experimental observations and finite element analysis outcomes. The findings indicated a strong alignment between the calculated values derived from the proposed strength estimation model, grounded in work principles, and the observed behavior evident in the test results.

**Keywords:** rotational friction; translational movement; strength estimation; finite element analysis; high-tension bolt; friction pad; work principal

## 1. Introduction

In Japan, a country prone to earthquakes, various technologies have been developed to prevent damage to buildings from seismic events. Vibration control aims to mitigate structural damage by absorbing seismic energy through energy dissipation systems, which are classified into four types: passive, active, semi-active, and hybrid [1–4]. Among these systems, various vibration control devices have been developed and categorized into types, such as friction, viscoelastic, and viscous fluid [1]. This research specifically focuses on friction dampers, a type of device investigated and developed by researchers and companies. Friction dampers absorb seismic energy through friction in translational or rotational movements. They are recognized for several advantages, including [5,6]:

- Large rectangular hysteresis loops and stable cyclic behavior.
- Independence of performance (frictional force obtained) from sliding speed and temperature.
- Energy dissipation through friction rather than the damping process of yielding.
- Relatively small and more compact size compared to other dampers.
- Simplicity and cost-effectiveness.
- Damage-proof operation during seismic events.

The application of friction dampers to building structures has been studied since the early 1980s [1]. Some incorporate a link mechanism in the brace, known as the Pall Friction Damper [7,8]. Additionally, an improved version of the Pall Friction Damper has been proposed to reduce manufacturing costs and enhance workability [9–12].

Linear (unidirectional) friction dampers have been studied since the 1980s [13,14]. In Japan, research and development of this type of friction damper have been conducted [15–17],

demonstrating a reduced seismic response with their use [18,19]. Practical applications of research results have been observed [18,20,21]. Takeuchi et al. proposed a new method to design a response control with optimal friction damper distributions based on the strength index and retrofit design method, confirming their validity [22,23].

While the linear friction damper allows translational movement on a friction surface, Mualla et al. proposed a rotational friction damper [24]. They conducted an experiment with a 5000 kN capacity rotational friction damper and performed Finite Element Analysis (FEA) investigations on it [25], demonstrating the application of high-capacity rotational friction dampers [26,27]. Ongoing research on novel rotating friction dampers is in progress, and the results are still being showcased. For example, Sui et al. investigated various types of steel friction pads for rotational friction dampers [28]. Javidan et al. [29,30] proposed a rotational friction damper joint with an eccentricity from the diagonal of the bay, demonstrating that structural elements remain elastic when using the damper and its application. Among the latest research findings, there is a self-centering rotational friction damper capable of reducing the residual displacement of structures [31,32] and a force-resisting rotary friction damper proficient in efficiently exerting frictional moments [33].

While rotational friction dampers are currently in practical use [5,8,18,20,21,26,27,30] and new dampers are still under development, the installation of these dampers in a building frame typically requires traditional pin joints with clevises or cylindrical hinge pins. Achieving optimal damper performance demands a minimal clearance between the pin and clevis, necessitating a level of manufacturing accuracy higher than that required for bolted joints. Consequently, the ultimate objective of our research is the development of a rotational friction damper that eliminates the need for pin joints. Significantly, this rotating friction damper possesses a unique feature, allowing not only rotation but also translational movement, a characteristic unprecedented in such devices. This represents the most pivotal feature of our damper. Unlike the rectangular hysteresis characteristics observed in existing friction dampers, which manifest as an increase in load with displacement or a hardening-like phenomenon, this feature sets our damper apart. Such a damper, endowed with this distinctive attribute, may find application as a fail-safe mechanism [34,35] in large earthquakes, presenting a promising avenue for future research.

The proposed rotational friction damper [36] comprises two components: one for rotation only and another for rotation with translational movement. Experimental tests were conducted to investigate the basic characteristics of the rotational friction damper with translational movement. Additionally, strength estimation models were proposed, and their validity was discussed in conjunction with test results and finite element analysis.

## 2. Proposed Rotational Friction Damper

The rotational friction damper, as proposed in our study, can be installed at the terminations of a brace within a building frame, illustrated in Figure 1. Figure 2 depicts a single unit of this damper along with its corresponding movement. Friction pads are positioned between the arm plate and the other plates and are subjected to surface pressure using high-tension bolts. The dynamics of the damper operation exhibit distinct characteristics: the upper side, referred to as 'R', experiences rotational forces exclusively, while the lower side, labeled 'RT', undergoes both rotational and translational movements, as demonstrated in Figure 2. Both 'R' and 'RT' function as friction dampers, obviating the necessity for pin joints at their connections.

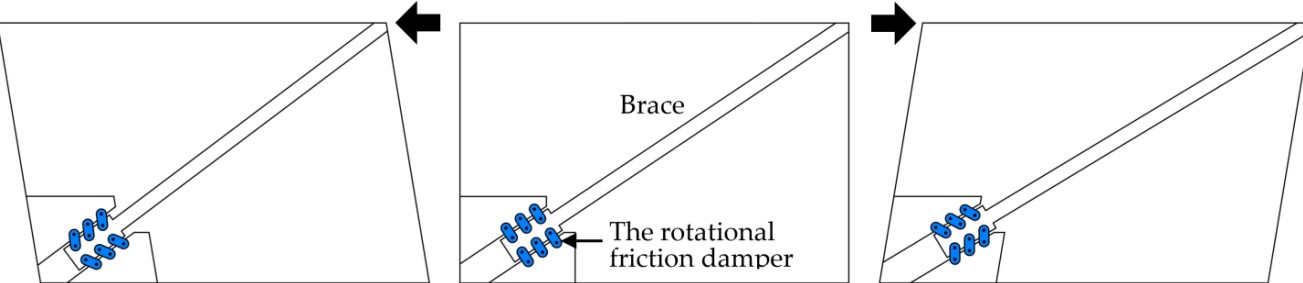

**Figure 1.** The damper in the frame and its behavior.

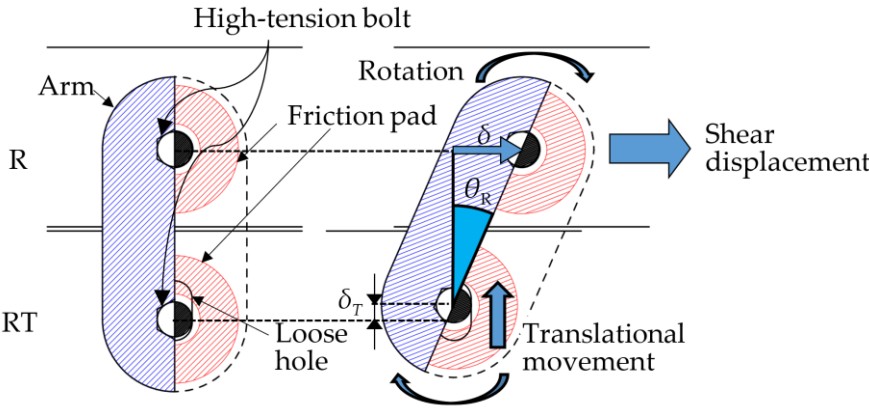

**Figure 2.** Working mechanism of the rotational friction damper (R and RT).

The strength of the damper is easily modifiable by adjusting the quantity of rotational friction dampers employed. Furthermore, control over the strength of these rotational friction dampers can be achieved by introducing axial force using high-tension bolts and adjusting the dimensions of the arm plates. Additionally, our proposed damper offers several advantages. While it can be manufactured in a factory, akin to conventional rotational friction dampers, our design allows for on-site connection to a brace and gusset plate using high-tension bolts, eliminating the need for pins and clevises. This could improve the construction management.

## 3. Specimen Configuration of the Rotational Friction Damper

Figure 3 illustrates the specimen configuration of the rotational friction damper. Figure 3b provides an internal view of the specimen, showcasing the absence of the arm and disc spring components. Friction pads are positioned between the arm plate and the center or loading plates and are subjected to surface pressure using high-tension bolts. The arm, center, and loading plates in the test specimen were constructed from stainless steel (SUS304) due to its inherent abrasion resistance. The friction pad assembly comprises a core plate (6 mm) wedged between two layers of friction material (1 mm each), as depicted in Figure 4.

As indicated by prior studies [22,23], the bolts were tightened to attain a standard force, resulting in an average surface pressure of approximately 10 N/mm² at the friction interface. To mitigate the reduction in surface pressure (bolt axial force) caused by alterations in the friction pad's thickness due to friction, disc springs were employed. Considering each disc spring's capacity of approximately 19 kN for the specimen, the configuration included eight disc springs aligned in the same direction. Furthermore, steel pipes were utilized to envelop the bolts, effectively reducing fluctuations in axial force stemming from shear forces acting upon the bolts.

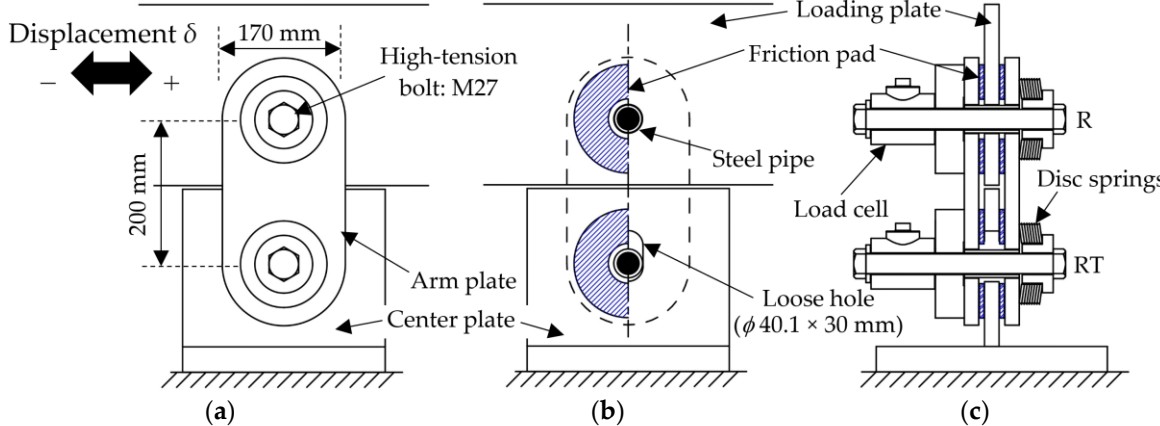

**Figure 3.** Specimen configuration. (**a**) Front elevation; (**b**) front elevation (inside); (**c**) side elevation (cross section).

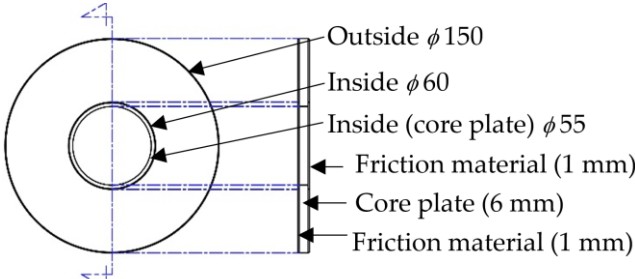

**Figure 4.** Configuration of friction pad.

The proposed rotational friction damper comprises two distinct components: a rotational friction damper (R) and a rotational friction damper with translational movement (RT), detailed in Figure 2. To individually analyze the behavior of R and RT, separate experiments were carried out for specimen R and specimen RT in this study. Consequently, a pin joint was employed at the RT side for specimen R, and conversely, at the R side for specimen RT, as illustrated in Figure 5.

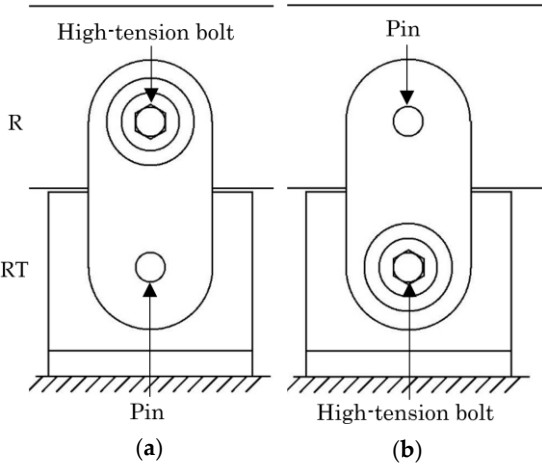

**Figure 5.** Specimen setup. (**a**) Specimen R; (**b**) specimen RT.

## 4. Experimental Tests

### 4.1. Testing Procedures

Figure 6 depicts the test setup of the specimen. Horizontal displacement was applied to the loading plate, which was dynamically executed via a 200 kN servo hydraulic testing

machine. The input waveform followed a sine wave pattern. Measurement parameters encompassed load, displacement along the loading direction, bolt axial force (measured using a load cell), and friction pad temperature. To confine the vertical displacement of the plate during experiments, a linear guide was integrated onto the loading plate.

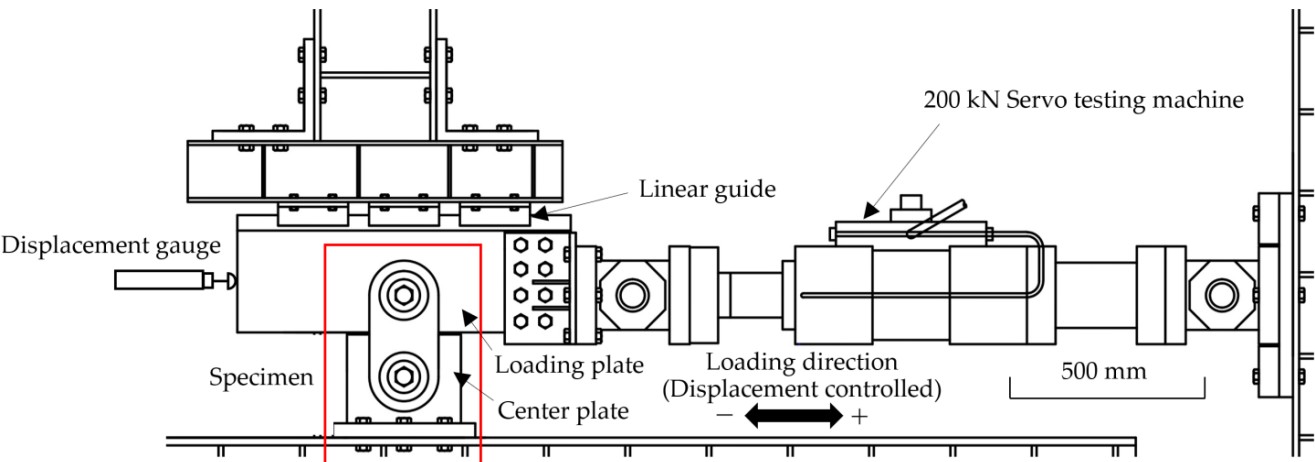

**Figure 6.** Experimental setup.

Table 1 presents the loading protocol established through reference to the prior literature on friction dampers [15–18,21,22]. The amplitudes were calibrated within the range of 10 to 80 mm, roughly equating to 1/400 to 1/50 of the inter-story deformation angle of the building. The experiments were consecutively conducted across the entirety of the loading protocol, denoted from No. 1 to No. 12, without re-tightening the high-tension bolts of the dampers.

**Table 1.** Loading protocol.

| Set No. | Loading Name | Amplitude (mm) | Frequency (Hz) | Cycles |
|---------|--------------|----------------|----------------|--------|
| 1 | S20–1 | 20 | 0.005 | 10 |
| 2 | S20–2 | 20 | 0.005 | 100 |
| 3 | D20–1 | 20 | 0.1 | 10 |
| 4 | D20–2 | 20 | 0.5 | 10 |
| 5 | D20–3 | 20 | 0.5 | 100 |
| 6 | S20–3 | 20 | 0.005 | 10 |
| 7 | D20–4 | 20 | 0.1 | 10 |
| 8 | D20–5 | 20 | 0.5 | 10 |
| 9 | D10–1 | 10 | 0.5 | 10 |
| 10 | D10–2 | 10 | 1 | 10 |
| 11 | D50 | 50 | 0.2 | 10 |
| 12 | D80 | 80 | 0.1 | 10 |

*4.2. Test Results*

Figure 7 illustrates the load–displacement relationship derived from the experiments, with loading protocols No. 8 (amplitude ± 20 mm) and No. 12 (amplitude ± 80 mm) serving as examples for R and RT, respectively. For both R and RT in No. 8, typical rectangular hysteresis loops, characteristic of friction dampers, are observed. Conversely, in No. 12, distinctive hysteresis loops are evident between R and RT. RT displays higher loads correlating with increased displacement due to translational movement, whereas R maintains a rectangular hysteresis loop. This distinction represents a key characteristic of the proposed rotational friction damper with translational movement. It is noted that R exhibits higher loads compared to RT at 0 mm displacement, attributable to variations in the friction coefficient.

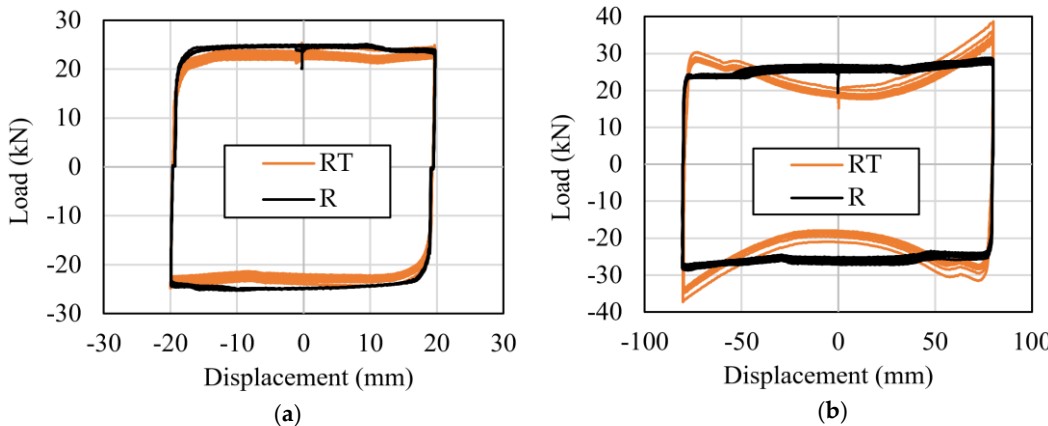

**Figure 7.** Load–displacement relationship. (**a**) No.8; (**b**) No.12.

Figure 8 shows the relationship between the friction coefficient and the cumulative number of cycles for R and RT, respectively. The friction coefficient values are extracted at 0 mm displacement within each cycle. The numerical labels 1 to 12 along the horizontal axis correspond to the loading protocol numbers specified in Table 1. In both instances of R and RT, a trend emerges where the friction coefficient experiences an initial increase for No. 1 and No. 2, reaching a near-stable value after approximately 60 cycles. Beyond No. 3, both R and RT exhibit a consistent friction coefficient around 0.3, albeit with some scattered values.

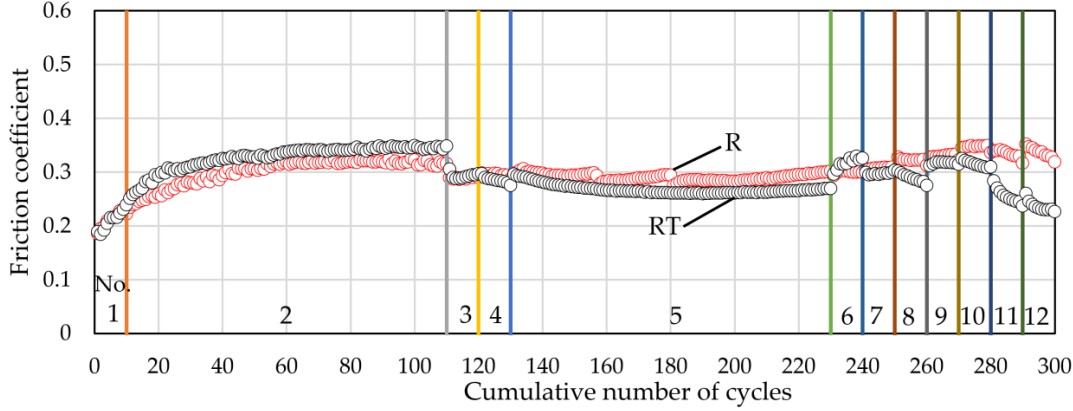

**Figure 8.** Behavior of friction coefficient.

## 5. Strength Estimation Model of Rotational Friction Damper

The proposed rotational friction damper experiences both rotation and translational movement simultaneously. While an equation has been proposed for scenarios involving rotation alone, denoted as R, it is noteworthy that in cases where translational movement (RT) is present, the hysteresis loop exhibits a different shape, as depicted in the test results (Figure 7). Consequently, for an accurate estimation of behavior and strength in the rotational friction damper, we introduced two strength estimation models (Model 1 and Model 2) in this study. Model 1 is rooted in the work principle, while Model 2 derives strength by separately calculating the contributions from the rotation and translational movement, summing these elements to determine the overall strength.

### 5.1. Model 1

The work principle was employed to estimate the strength of a rotational friction damper exhibiting translational movement. The strength was derived from the equilibrium of two components: $W_O$ and $W_I$. Here, $W_O$ represents the work resulting from the load, $P$, and the horizontal displacement of the specimen, $\delta$, while $W_I$ pertains to the work

attributed to the frictional force and the motion of the friction pad. Each component of work is delineated below.

### 5.1.1. Work $W_O$

The work, $W_O$, resulting from the load, $P$, and the horizontal displacement, $\delta$, of the specimen can be calculated using the following equation:

$$W_O = P \cdot \delta \tag{1}$$

### 5.1.2. Work $W_I$

The work, $W_I$, can be expressed as the product of the displacement in each region and the frictional force. Therefore, $W_I$ can be calculated by the following equation:

$$W_I = \sum_{i=1}^{n} \mu N_i \Delta l_i \tag{2}$$

Here, $\mu$ represents the friction coefficient, $N_i$ is the axial force for each divided region, and $n$ indicates the number of friction pad divisions. $\Delta l_i$ signifies the displacement of each area of the friction pad.

Figure 9 illustrates the displacement ($\Delta l_{1\sim4}$) of the center of each region, as an example in the case where $n = 4$.

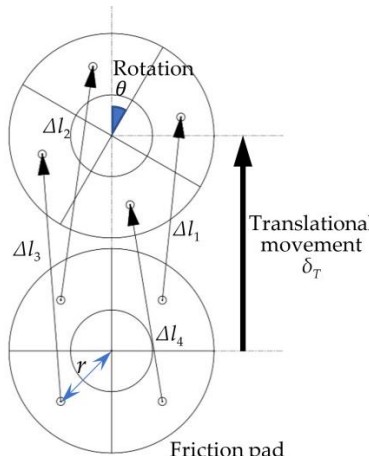

**Figure 9.** A model of friction pad movement (in the case of $1 \times 4$).

The friction surface area was segmented both radially and circumferentially, resulting in five different segmentation patterns explored in this study (illustrated in Figure 10). The work was calculated for each of these five patterns.

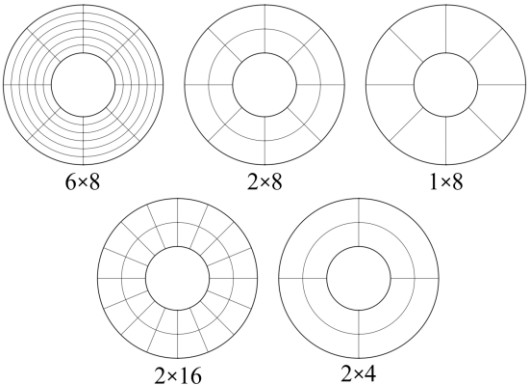

**Figure 10.** Patterns of the friction surface segmentation.

For test R, wherein the friction pad experiences only rotational movement without any translational movement, segmentation in the circumferential direction becomes unnecessary. Consequently, segmentation of the pad itself is no longer required for the R specimen.

### 5.1.3. Calculation of Strength for Both R and RT

Since the work $W_O$ of the external force and the work $W_I$ of the internal force are balanced, the load $P$ is obtained from Equations (1) and (2) and is as follows:

$$P = \sum_{i=1}^{n} \frac{\mu N_i}{\delta} \Delta l_i \tag{3}$$

### *5.2. Model 2*

### 5.2.1. Calculation of Strength for R

In Model 2, the strengths attributed to rotational and translational movements were computed independently. To determine the strength of the rotational friction damper, it is essential to ascertain the rotational friction moment, $M_R$ [13]. The integration of the rotational friction moment, $dm$, across small sections of the friction pad enables the calculation of the rotational friction moment, $M_R$, assuming a consistent surface pressure of the friction pad. When $N$ represents the normal pressure generated by bolts at any location between the friction pad and the center or loading plates, the resulting frictional force acting on an elemental area can be expressed as $\mu N dA$, where $\mu$ is the friction coefficient and $dA$ is the area ($rdrd\theta$) of the element. The moment generated by this elemental friction force around the center of the friction pad is $\mu N r dA$. Therefore, the total moment is obtained by integrating over the area, resulting in $\int \mu N dA$. Consequently, the frictional moment can be calculated using Equation (4):

$$\begin{aligned} M_R &= \int_A dm = \int_0^{2\pi} \int_{r_1}^{r_2} \mu \cdot \frac{N}{A} \cdot r^2 dr d\theta \\ &= \frac{2}{3} \cdot \mu \cdot N \cdot \frac{r_2^3 - r_1^3}{r_2^2 - r_1^2} \end{aligned} \tag{4}$$

Figure 11 shows the geometric parameters of the friction pad. $r_1$ = 30 mm and $r_2$ = 75 mm according to the size of the friction pad, as shown in Figure 4.

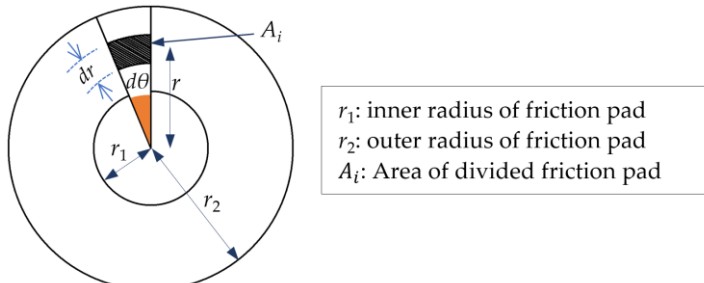

$r_1$: inner radius of friction pad
$r_2$: outer radius of friction pad
$A_i$: Area of divided friction pad

**Figure 11.** Geometric parameters of the friction pad.

The strength of the rotational friction damper can be calculated as follows:

$$P_R = \frac{M_R}{200 - \delta_T} \tag{5}$$

$$\delta_T = 200\{1 - \cos(\theta_R)\} \tag{6}$$

$\delta_T$ represents the vertical displacement, as depicted in Figure 2, where the initial span of 200 mm is reduced in accordance with $\delta_T$. For reference, Figure 12 illustrates the correlation between $\delta$ and $\delta_T$.

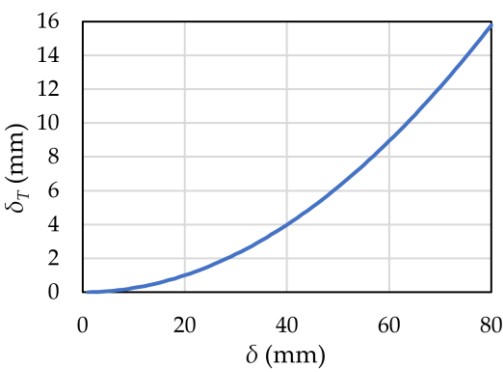

**Figure 12.** Translational movement of high-tension bolts due to specimen displacement.

### 5.2.2. Calculation of Strength for RT

Since rotation occurs simultaneously with translational movement as demonstrated in Figure 13, the rotational friction moment, $M_{RT}$, accounting for translational movement, is calculated using Equation (7), where *n* represents the number of divisions of the friction pad. $P_{RT\_R}$ is derived from the obtained rotational friction moment, $M_{RT}$, utilizing Equation (8):

$$M_{RT} = \sum_{i=1}^{n} \mu \cdot N \cdot \frac{A_i}{A} \cdot r \sin\left(\theta_{RTi} - \theta_R\right) \tag{7}$$

$$P_{RT\_R} = \frac{M_{RT}}{200 - \delta_T} \tag{8}$$

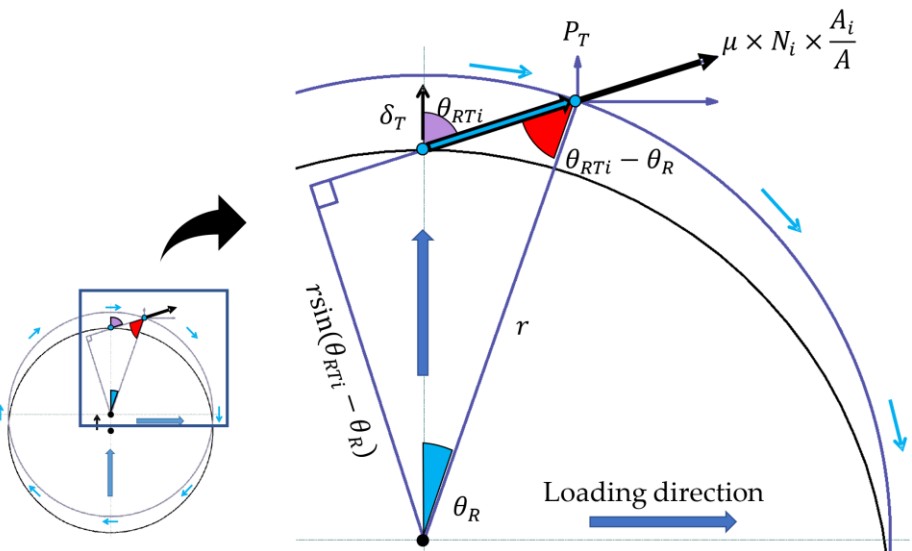

**Figure 13.** Example of friction pad behavior.

The translational load, $P_{RT\_T}$ (load acting in the translational direction), can be calculated through Equation (9). Subsequently, the load, $P_{RT}$, for the rotational friction damper with translational movement, is determined using Equation (10):

$$P_{RT\_T} = \sum_{i=1}^{n} \mu \cdot N \cdot \frac{A_i}{A} \cdot r \cos(\theta_{RTi}) \tag{9}$$

$$P_{RT} = P_{RT\_R} + P_{RT\_T} \cdot \tan(\theta_R) \tag{10}$$

*5.3. Comparison of the Estimation Models*

In Figure 14, the load–displacement curves are depicted to analyze the effects of segmentation on Model 1's strength. Regarding the radial segmentations in Figure 14a, the $1 \times 8$ segmentation exhibits a slightly smaller load compared to the others, while both the $2 \times 8$ and $6 \times 8$ segmentations demonstrate almost identical behavior. This suggests that two segments in the circumferential direction suffice for effective segmentation. As for the circumferential segmentations illustrated in Figure 14b, Load-displacement relationships are almost exactly the same. Hence, it is evident that a $2 \times 4$ segmentation would be a reasonable choice for calculations.

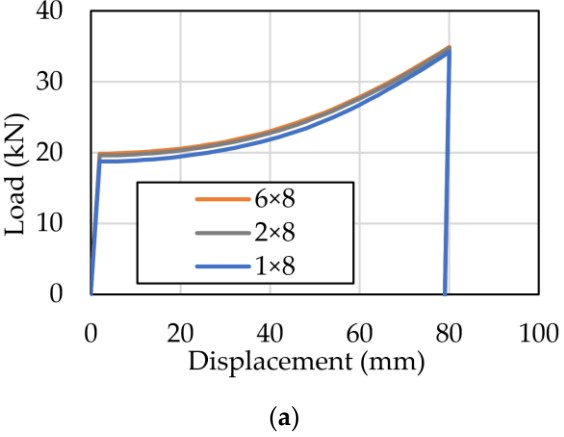
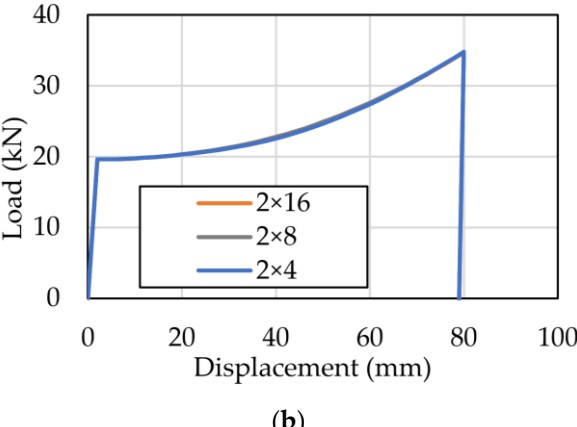

(**a**)   (**b**)

**Figure 14.** Effect of segmentations for Model 1. (**a**) Effect of radial segmentation; (**b**) effect of circumferential segmentation.

Figure 15 presents a comparative analysis of the strength estimation models for both R and RT, employing a $2 \times 4$ segmentation, as indicated in Figure 14. The load–displacement curves for Models 1 and 2 demonstrate close agreement in both R and RT scenarios. In R (a), the load remains relatively constant due to the consistent nature of the rotational friction moment. However, a slight increase in load is noticeable in R, attributed to the reduction in the distance (span) between the arm holes caused by translational movement, as depicted in Figure 2. In contrast, in RT (b), the load exhibits an increase corresponding to the displacement resulting from translational movement. Models 1 and 2 present nearly identical results across the range of observed displacements.

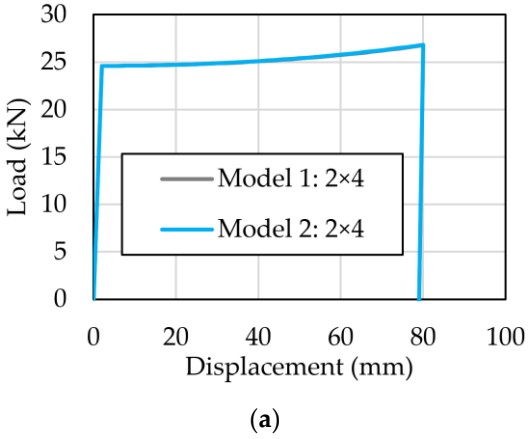
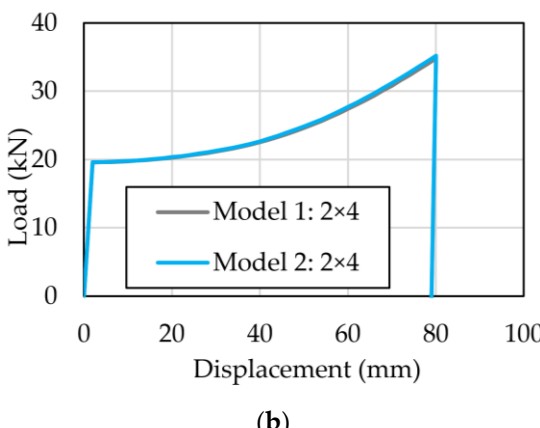

(**a**)   (**b**)

**Figure 15.** Comparison of Model 1 and Model 2. (**a**) R; (**b**) RT.

To reduce the number of segmentation counts for Model 1, adjustments were made to the measurement point of $\Delta l$, as depicted in Figure 16. The modified measurement point

for Model 1 g in Figure 16b now represents the center of gravity within the segmented region, whereas the original measurement point for Model 1 is displayed in Figure 16a.

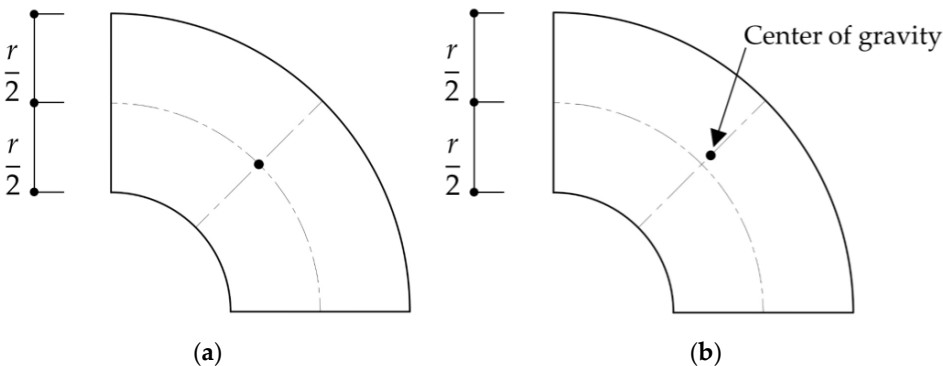

(**a**)  (**b**)

**Figure 16.** Measurement point of Δ*l* for Model 1 and Model 1 g. (**a**) Model 1; (**b**) Model 1 g.

According to Figure 17, while the 1 × 4 segmentation of Model 1 exhibits a slightly lower load, the behavior of the 1 × 4 segmentation in Model 1 g closely resembles that of the 2 × 4 segmentation in Model 2 × 4. As a result, based on the insights from Figure 17, we intend to adopt Model 1 g (1 × 4 segmentation) as the primary strength estimation model for discussion purposes.

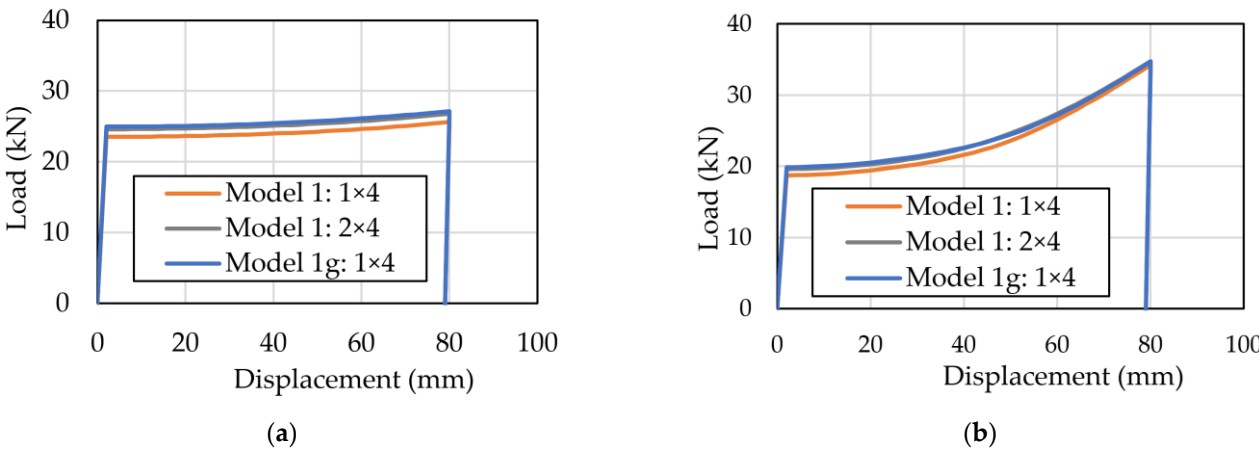

(**a**)  (**b**)

**Figure 17.** Comparison of Model 1 and Model 1 g. (**a**) R; (**b**) RT.

## 6. Validity of Strength Estimation Model

For the validation of the strength estimation model (Model 1 g: 1 × 4 segmentation), the load–displacement curves of the model are compared against those derived from both experimental tests and Finite Element Analysis (FEA).

### 6.1. FEA

In order to simulate and replicate the behavior of the proposed rotational friction damper, Finite Element Analyses (FEA) were conducted using MSC Marc 2022. The results were evaluated through a comparative analysis between the historical characteristics of experimental tests and the analytical findings.

The analytical model, as illustrated in Figure 18, was configured to mirror the specimen's dimensions and layout. An eight–node reduced integration element type was utilized, considering the pins as rigid. Standard material properties were employed for the specimen. The number of elements used was 620 for friction pads, 2532 for arms, and 8169 for bolts, while the number of nodes was 480 for friction pads, 2136 for arms, and 1698 for bolts. The specimen elements maintained a minimum dimension of 0.74 mm × 0.91 mm × 5.00 mm.

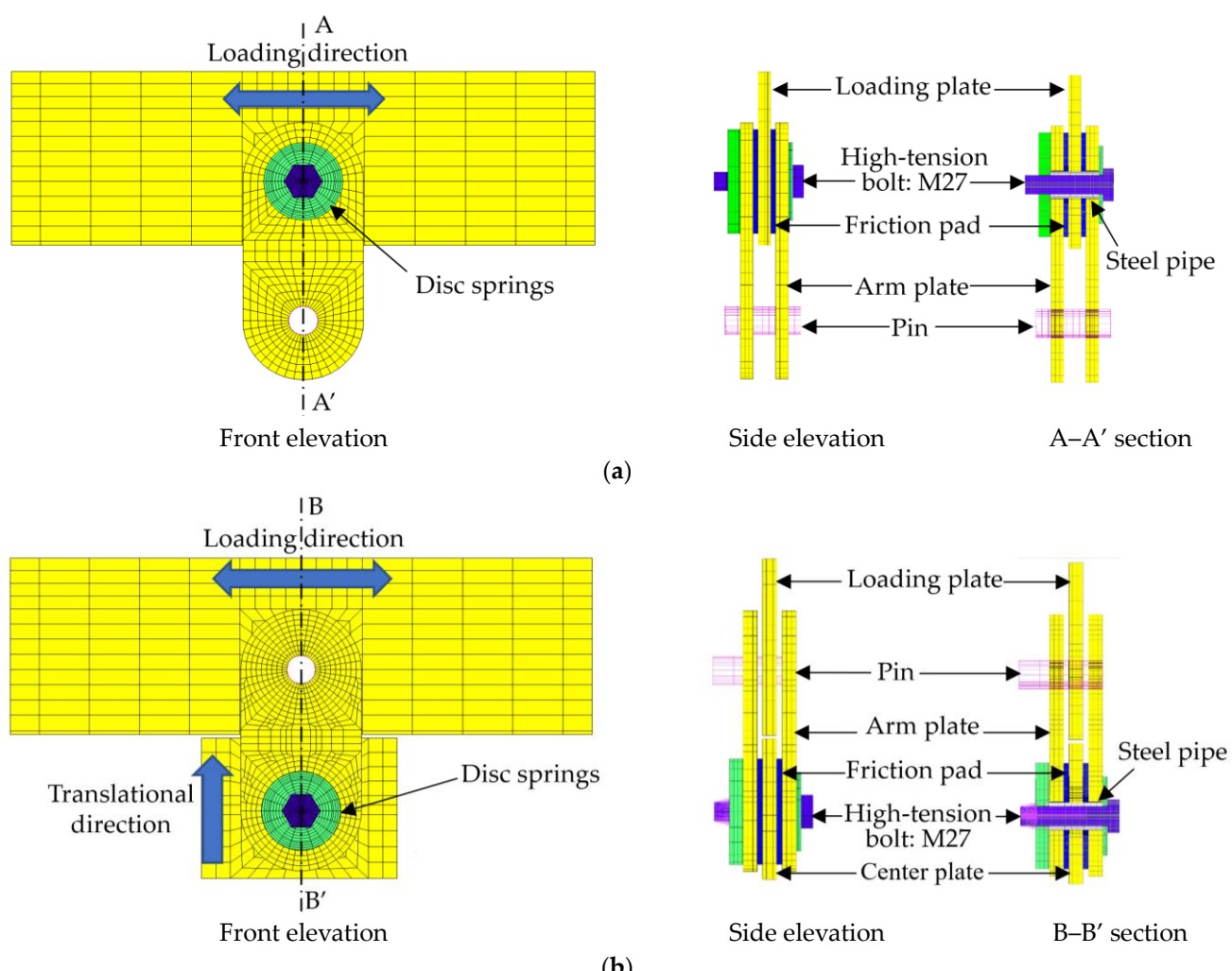

**Figure 18.** FEA models. (**a**) Model R; (**b**) Model RT.

Two analytical models were created to align with the test specimen: model R for rotation only and model RT for both rotation and translational movement. Contact friction was modeled using the Coulomb model, and the friction coefficients, along with the bolt axial forces, were set based on experimental test data. The friction coefficients for loading protocol No. 12 were set at 0.369 for R and 0.298 for RT. Additionally, the bolt axial forces were defined as 137 kN for R and RT, respectively, as derived from the test results. All components within this analysis were subjected to elastic stress.

*6.2. Comparison of Strength Estimation Model with Experimental Test and FEA Results*

Figure 19 depicts the hysteretic curves derived from Finite Element Analysis (FEA), experimental tests, and the strength estimation model (Model 1 g). The FEA results exhibit a close resemblance to the test results for both R and RT. Furthermore, the FEA successfully reproduces the load increment with displacement, mirroring the behavior observed in the strength estimation model for RT. The hysteretic curves of the strength estimation model, based on the work principle, exhibit a strong agreement with the experimental test results for both R and RT. This suggests that the proposed strength estimation model, rooted in the work principle, effectively simulates the hysteretic curves of the rotational friction damper.

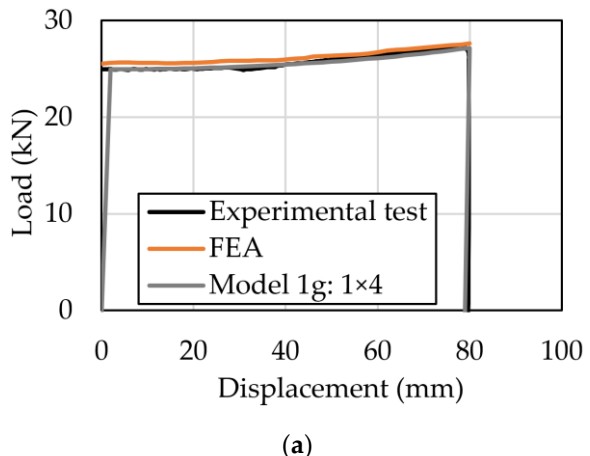 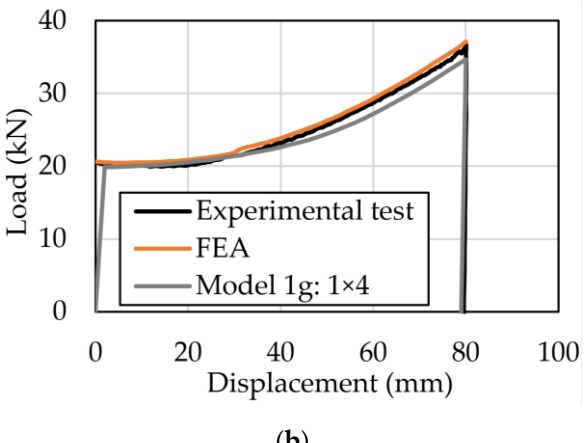

(**a**)  (**b**)

**Figure 19.** Load–displacement relationship. (**a**) R; (**b**) RT.

It is worth noting that in this field it is important to consider the dissipated energy. Therefore, we plan to address it in our future research.

### 7. Conclusions

Experiments involving the proposed rotational friction damper, considering translational movement, were conducted. Subsequently, a strength estimation model, rooted in the work principle, was developed for the rotational friction damper. This model was then comprehensively compared against both the experimental test outcomes and the results obtained from Finite Element Analysis (FEA). Key observations include:

- The rotational friction damper with translational movement, RT, demonstrates a load increase proportional to displacement, attributed to the effect of translational movement. In contrast, the rotational friction damper (rotation only), R, maintains a nearly constant load.
- The load–displacement curves of the strength estimation model, rooted in the work principle, exhibit a remarkable alignment with the experimental test results for both R and RT.

**Author Contributions:** Conceptualization, T.I.; methodology, S.T. and T.I.; validation, S.T. and T.I.; analysis and investigation, S.T. and K.I.; writing—original draft preparation, S.T.; writing—review and editing, All authors; supervision, K.G. and M.Y. All authors have read and agreed to the published version of the manuscript.

**Funding:** This research was funded by The Japan Iron and Steel Federation.

**Data Availability Statement:** The data in this study are available on request from the corresponding author.

**Acknowledgments:** The authors wish to thank the staff of the Technical Center at National Institute of Technology, Ariake College for their help with the experiments.

**Conflicts of Interest:** The authors declare no conflict of interest. The funders had no role in the design of the study; in the collection, analyses, or interpretation of data; in the writing of the manuscript; or in the decision to publish the results.

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
