# Peer review of "Strength Estimation and Fundamental Characteristics of the New Rotational Friction Damper with Translational Movement"

_machines, doi:10.3390/machines12010015_

Round 1
Reviewer 1 Report
Comments and Suggestions for Authors
Overall, a well done and written article. I suggest minor corrections.
First, please add nomenclature.
Please explain what dm means in Eq. (4). I don't understand why in Eqs. (7, 9, 10) multiplication is marked with an "x".
I suggest adding information in Chapter 6 on the number of elements and nodes in the FE model.
I suggest supplementing the references with more English-language titles from other scientific centers.
Reviewer 2 Report
Comments and Suggestions for Authors
The paper addresses a very important and interesting topic on a friction damping mechanism in beam structures. Approaches for evaluating the damping properties as a function of the kinematics of translation and rotation are presented, evaluated and compared with FE calculations.
The paper is interesting and written in a mostly understandable way, but I have a few recommendations and points that I think require further explanation.
1) The text should be proofread by a native English speaker.
2) Figure 1 does not clearly show how the damper actually works, as does figure 2. As far as I understand it, the damping is to be generated by friction pads, which are either not shown or not labeled.
3) My biggest problem is the actual procedure. As far as I understand it, the main issue is that the simultaneous presence of translation and rotation makes it difficult to specify a resultant force/moment. From my point of view, it would be more appropriate to set up a kind of measure for the damping power (or dissipated energy), in which the actual velocity field is first set up as a superposition of translation and rotation and then used in the integral. In principle, this is similar to equation 4, except that the individual powers or energies are integrated rather than the moment. From my point of view, this should be possible in an analytically way and be implemented in a scientific paper.
4) It is unclear to me why the FE calculation should be worse than the selected simple model. In Figure 19b, the curve in the FEA solution shows unsmoothness that looks very much like numerical errors that could be prevented by a finer mesh.
Comments on the Quality of English Language
The paper should be proofread by an English native speaker.
Round 2
Reviewer 2 Report
Comments and Suggestions for Authors
The authors have addressed and implemented my comments and suggestions. I am still of the opinion that the derivation of the strength measure is not consistent because the velocity field prevailing in the contact, which is composed of translation and rotation, contradicts the calculation in equation (4), in which it is assumed that all local forces act tangentially (only rotation). So for me it is not a real model but actually just a heuristic formula that seems to have a good agreement with measurements.
Since adding my comment to the paper would be too time-consuming, I recommend accepting it as is. Perhaps the authors will go into this in more detail in future papers.